# Three-dimensional shape cues affect human and artificial recognition systems differently

**Mikayla Cutler**[1☯], **Luke Baumel**[1☯], **Joseph Tocco**[1], **William Friebel**[1],
**George K. Thiruvathukal**[2], **Nicholas Baker**[1*]

**1** Department of Psychology, Loyola University Chicago, Chicago Illinois, United States of America,
**2** Department of Computer Science, Loyola University Chicago, Chicago Illinois, United States of America

☯ These authors contributed equally to this work.
* nbaker1@luc.edu

## Abstract

Humans and neural networks use shape and texture information differently. While humans weigh shape heavily in their ultimate classification decision, neural networks are more biased towards texture cues. Many tests of shape vs. texture bias have focused on shape recognition from an object's external contour. However, shape information is also conveyed through internal contours, shading, and attached shadows, especially when an object is viewed from noncanonical perspectives. Using models from ShapeNet, we created datasets of 120,000 texture-substituted images of objects from many viewpoints with and without shading and attached shadows. We tested humans' and several neural networks' ability to classify these objects by both their shape and their texture. Humans were much better at classifying texture-substituted objects by their shape than any network, although these differences were greater when shape was defined only by the external contour than when 3D cues were included. Our findings suggest that networks' texture bias is reduced when 3D cues are included in images. We next tested whether the inclusion of 3D cues benefitted humans and neural networks more for images of objects viewed from canonical or noncanonical perspectives. Consistent with earlier research, we found that 3D cues primarily benefitted humans for noncanonical images. For neural networks, the greatest performance gains were for canonical images. These findings suggest fundamental differences in how humans and networks use shading and attached shadows for object recognition. We argue that humans use these cues to infer objects' 3D structures while neural networks use them as another surface-level cue like texture.

## Introduction

Deep neural networks (DNNs) match human performance on a variety of visual tasks [1–8] and predict neurophysiological activity in the visual brain [9–14]. These successes have stirred interest in networks' usability as image-computable models of

**Data availability statement:** All relevant data for this study are publicly available from the Zenodo repository (https://doi.org/10.5281/zenodo.20073242).

**Funding:** This work was funded by a Carbon Fellowship to LB (https://www.luc.edu/sustainability/research/studentresearchopportunities/). The funders had no role in study design, data collection and analysis, decision to publish, or preparation of the manuscript.

**Competing interests:** The authors have declared that no competing interests exist.

visual perception. However, there are several key differences in the way humans and deep networks process visual information that have raised caution about their plausibility as models of visual cognition [15–20].

Among the most fundamental differences between humans and deep networks is in their use of shape and texture. For humans, both shape and texture play a key role in object perception in the ventral visual pathway [21–25]. Texture is processed earlier [25], and some work suggests that for "ultrarapid" object recognition, texture plays a primary role in object recognition [26,27], although it has also been argued that the ventral stream's rapid processing of texture is in service of "de-texturizing" the image through contextual modulation to segment an object's shape from its background [25]. When shape and texture are disentangled, humans are biased towards shape [22,28–32]. Neural networks, meanwhile, are more reliant on the texture of the object for categorization [33–39]. The inclusion of deceptive texture hurts DNN object recognition considerably more than human recognition [34,35,37]. Networks' texture bias has been reduced by alternative training that limited the diagnostic value of texture information [33,35,37,39,40] or boosted the shape signal [36]. Whether these models leverage visual information like humans remains uncertain [41,42].

In studies where shape and texture information are put in direct competition with each other, a common approach is to remove the texture from an image of the original object and then substitute the texture from a different object onto the original object [34,35]. Texture substitution at the image level can be done by converting color images of an object to grayscale and taking the elementwise product of the grayscale image and a texture image (Fig 1a). Texture substitution can also be done by converting the object to a binary silhouette and substituting the figural region with a different texture (Fig 1b). The former approach does not fully remove texture cues from the object's shape, as can be seen from the preserved features in the eyes and face of the cat in Fig 1a, which are the result of grayscale texture intensity [81]. The latter approach removes all texture but also any 3D shading information contained within the shape's bounding contour.

Because of limitations in these approaches, it remains unstudied how shading and *attached shadows* (i.e., shadows cast from one part of an object onto another part of the same object) affect object recognition in DNNs. If networks recognize shapes more easily when shadows are preserved, one possibility is that their texture bias is caused not just by a preference for texture information but also by the loss of important shape 3D information conveyed by shading and attached shadows. Another possibility is that shading and attached shadows are treated as a texture cue by deep networks and are not leveraged by DNNs to infer objects' 3D structures.

These two possibilities have also been considered for humans' use of shading and attached shadows for object recognition. A great deal of information can be inferred about an object's 3D structure based on shading and shadows [43,44], and humans may make use of these cues to form structural representations of objects [43,45–47]. On the other hand, it has been argued that shadows are primarily used for recognition as an image-level cue, not for the inference of 3D structure [48, 49].

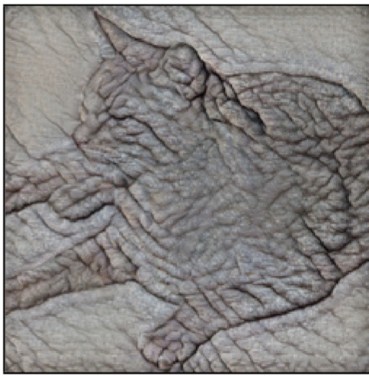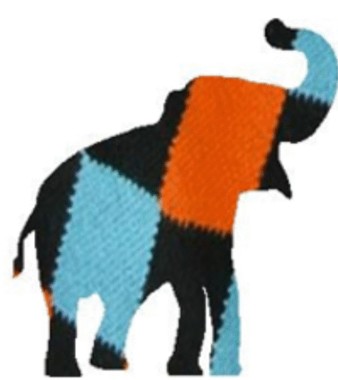

**Fig 1. Texture substitution examples. a)** Cat shape substituted with elephant texture. Reprinted from [35] under a CC BY license from **R.** Geirhos, 2018. **b)** Elephant shape substituted with argyle sock texture. Reprinted from Baker et al. (2018) under a CC BY license from *PLOS Computational Biology*, 2018.

We tested the interaction between shading and texture in both humans and DNNs by creating a novel dataset from 3D ShapeNet models overlaid with nondiagnostic texture. This allowed us to systematically manipulate texture information while preserving objects' shape and attached shadows. We first compared humans' and neural networks' shape vs. texture bias with and without shading or attached shadows, which we refer to as *3D cues*. We predicted that humans, who are highly sensitive to the bounding contour of objects [28,50], would benefit only marginally from the inclusion of 3D cues, but that DNN performance would improve substantially with 3D cues included.

To the extent that shadows do benefit shape recognition in humans and deep networks, we also sought to understand why they are beneficial. Do shadows aid recognition because they increase the image-level similarity between a novel instance of an object and previously seen images, or do they allow for inference about an object's 3D structure?

We tested this by comparing the performance advantage conferred by shadows for objects viewed from a typical (*canonical*) perspective vs. objects viewed from an atypical (*noncanonical*) perspective. In human perception, object recognition is easier for objects viewed from canonical perspectives [51–54]. Differences in recognition performance between canonical and noncanonical images are greater when the images include only the object's external contour than when internal contours [52,55] or 3D cues such as shading and attached shadows are also present [51,53].

Likely, the interaction between canonicality and the inclusion of internal contours and/or shading and attached shadows has to do with the formation of structural, volumetric representations of object shape. If deep neural networks benefit from the inclusion of 3D cues equally for canonical and noncanonical images, that would be evidence that they do not form volumetric shape representations like humans do. It would also demonstrate that the asymmetrical benefit of 3D information for canonical and noncanonical images observed in humans is inconsistent with the view that humans recognize objects based on purely image-level similarities. DNNs constitute an ideal observer model for this kind of object recognition, so differences between humans and neural networks imply a different computational strategy for object recognition.

## Materials and methods

### Models tested

We tested three state-of-the-art network architectures, ResNet-50, ViT-16, and SWIN. Below, we briefly describe key differences between the three models.

## ResNet-50

ResNet-50 [56] is a very deep convolutional neural network. While it functions like other deep convolutional networks, it includes "skip connections", which makes it possible for the network to include many layers while preserving the error signal in gradient descent. As one of the most famous convolutional networks, ResNet has been trained with several alternative curricula in efforts to increase its shape bias. We tested ResNet with three training methods.

1. ImageNet: The most standard curriculum for DNNs is ImageNet [57], a database of 1.2 million natural images of objects from 1,000 categories.

2. ImageNet and Stylized ImageNet: An alternative training method in which DNNs are trained on both standard ImageNet photographs and photographs from ImageNet that have undergone "style transfer", which converts them into an artist's painterly style. Doing so reduces the diagnostic value of texture and increases shape bias on some classification tasks [35].

3. Strong-blur: Another alternative training method aimed at simulating the visual experience of babies and vision in the periphery, two hypothesized causes of humans' shape biases. Blurred ImageNet images were convolved with a Gaussian blur kernel of varying sizes ($\sigma = 0, 1, 2, 4$ or $8$ pixels, with equal probability). Shape bias was significantly greater in ResNet models trained with blurring than models trained only on the standard ImageNet dataset [39].

## ViT-16

One of the first, and still among the most successful, transformer architectures applied to the object recognition task, ViT-16 [58] uses a multi-head mechanism to learn the complex interactions between 16x16-pixel image patches for image classification. ViT greatly outperforms ResNet on shape classification tasks absent any diagnostic texture [42], possibly because its self-attention mechanism allows it to learn long-range relations between parts of a shape.

Whereas convolutional networks are trained to learn local relations between nearby pixels in early layers, ViT is free to learn pixel relations between any patches in the image, regardless of distance. As a result, ViT needs much more training data than ReNet or SWIN (see below), but it performs very well when trained on large datasets. The model we tested was trained on JFT-300M, a large proprietary labelled image database, and finetuned on ImageNet for the 1,000 object categorization task.

## SWIN

We also tested SWIN [59], a hierarchical transformer architecture that breaks images into small 4x4 pixel image patches in early layers but merges them into successively larger patches in deeper layers of the network. SWIN uses a local self-attention head which allows relations to be learned only between pairs of image patches in the same window of attention. This window of attention convolves over the whole image to learn different pixel relations. SWIN has more inductive biases than ViT and therefore requires less training data to accurately classify images. The model we tested was trained on ImageNet.

## Stimuli

### 3D shape models

We selected 100 3D models from ShapeNet [60], a dataset of 3D shapes of both biological and non-biological objects. We chose ten object categories—five biological and five nonbiological—that were among the trained categories in ImageNet and had numerous high-quality models in ShapeNet. For each category, we selected 10 models. The biological categories were fish, elephant, butterfly, bird, and bear, and the non-biological categories were helmet, mailbox, bathtub, mug, and phone.

*Texture images* For each object category, we also found 10 high-resolution texture images to pair with the shapes. Texture images were cropped so that they included no background pixels.

*Testing datasets* We created two datasets, one 3D and one 2D, upon which to test both humans and DNNs. We generated these datasets by first taking photographs of each of the 100 3D shape models twelve times in 30° increments around the 3D model for a total of 1,200 model images (100 shapes x 12 orientations). All models were photographed with white surfaces on a black background.

For the 3D stimuli, we created 100 retextured images for each model image by multiplying the grayscale pixel values of the image with the RGB values of each of the 100 texture images. We created stimuli for the 2D dataset by converting the same photographed images into binary images and multiplying each pixel value (one for the figure or zero for the background) with a texture image. This resulted in 120,000 3D and 2D retextured images that differed only in their inclusion of shading and attached shadow cues.

We hand-coded each object in our database as being viewed from a canonical perspective or a noncanonical perspective. Researchers have debated what determines whether a viewpoint results in a canonical or noncanonical image of an object. One view is that canonicality depends primarily on the visibility of features typically associated with a particular object, such as an elephant's trunk or a piano's keyboard [54]. Others have argued that canonicality depends only on the familiarity of a viewpoint [61], although experiments where familiarity was controlled still showed canonicality effects [62]. For our images, which are of familiar objects, these competing ideas make similar predictions. Objects viewed from a canonical perspective faced toward the viewer or in profile to the viewer. Objects viewed from a noncanonical perspective faced away from the viewer and often self-occluded distinctive features for recognition. An example canonical and noncanonical image from the 2D and 3D datasets is shown in Fig 2.

## Network experiments

We tested each DNN on both of the datasets described above. The 1,000 object categories networks are trained on with ImageNet are more granular than the entry-level categories we used to create our testing sets. We identified the subcategories in ImageNet that belonged to each entry-level category. The subcategories associated with each entry-level category are shown in Table 1. To determine networks' classification decisions, we took the sum of all softmax values of subcategories that belonged to each entry-level category [63]. The networks' classification decisions were taken to be whichever entry-level category had the highest summed probability.

## Human experiments

We conducted two human experiments, one which compared humans' sensitivity to shape and texture, and the other which compared humans' ability to classify images' shapes when objects were viewed from canonical and noncanonical perspectives.

## Participants

125 undergraduate students from Loyola University Chicago ($M_{age}$ = 18.5) participated in both experiments for course credit. An additional six undergraduates participated only in Experiment 2 (125 total participants for Experiment 1 and 131 total participants for Experiment 2). All participants had normal or corrected-to-normal vision and were naïve to the purpose of our experiments.

Data collection took place between 22/09/2025 and 24/02/2026. All participants gave written consent before participating in the experiments.

## Display and apparatus

Participants were seated approximately 70 cm away from a 60 cm MSI Optix G2412 computer monitor. The monitor's resolution was set to 1920x1080 pixels and had a refresh rate of 144 hz.

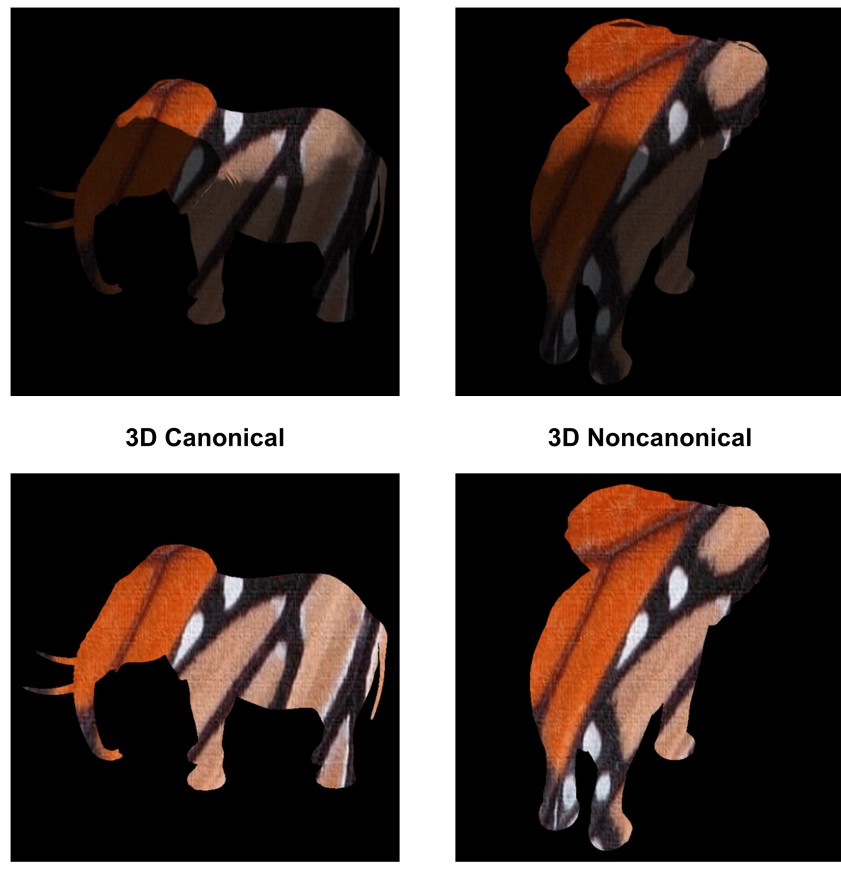

**3D Canonical**  **3D Noncanonical**

**2D Canonical**  **2D Noncanonical**

**Fig 2. Sample images from Experiments 1 and 2.** Top row: An elephant-butterfly with 3D information viewed from a canonical and noncanonical perspective. Bottom row: The same elephant-butterfly with only 2D information.

**Table 1. Mapping of ImageNet categories to the ten entry-level categories used in our experiments.**

| | |
|---|---|
| **Bathtub** | Bathtub, tub/vat |
| **Bear** | Koala, brown bear, black bear, polar bear, sloth bear, red panda, giant panda |
| **Bird** | House finch, snowbird, indigo finch, robin, bulbul, jay, magpie, chickadee, water ouzel |
| **Butterfly** | Ringlet butterfly, monarch butterfly, cabbage butterfly, sulphur butterfly, lycanid butterfly |
| **Elephant** | Indian elephant, African elephant |
| **Fish** | Goldfish, crampfish, devilfish, eel, coho, rock beauty, anemone fish, sturgeon, garfish, lionfish, pufferfish |
| **Helmet** | Crash helmet, football helmet, gasmask |
| **Mailbox** | Mailbox/letter box |
| **Mug** | Coffee mug |
| **Phone** | Cellular telephone, dial telephone, payphone |

## Experiment 1

In Experiment 1, we used a 2 (recognition cue) x 2 (presence of 3D cues) design to test participants' ability to recognize objects by their shape vs. texture with or without shading and attached shadows consistent with the object's shape.

We manipulated the recognition cue (shape or texture) in two trial blocks. In the first 160 trials, they were instructed to identify an object's shape from 10 categories, irrespective of its texture. We used 80 3D and the same 80 2D retextured images, which were randomly interleaved. Eight shapes from each category were randomly selected with a random texture and orientation with the caveat that the image's texture and shape could never belong to the same category. Each object category was bound to a single key, and participants were instructed to press the key to identify the shape shown in the presented image. They were instructed to respond as quickly as possible without compromising accuracy. They completed 10 practice trials to familiarize themselves with the instructions and the mapping between object categories and response keys before beginning the main experiment.

Once they had completed the shape trials, participants were given a new set of instructions in which they were told to identify the texture belonging to the object in a presented image. As in the shape trials, stimuli were selected from the 2D and 3D retextured datasets, randomly interleaved and in equal proportions. Eighty (eight textures for each of the 10 categories) images were selected from each dataset with shape randomly selected from the other nine categories and random orientation. Participants completed 10 practice trials before beginning the main experiment.

## Experiment 2

In Experiment 2, participants' only task was to identify the shape of the presented object, irrespective of texture. We had two independent variables, each with two levels.

The first factor we manipulated was shading cues. Images were either shaded by the three-dimensional structure of the object and had attached shadows (*3D*) or had no shading or attached shadows (*2D*). The second factor was canonicality of viewpoint. Images could be viewed from either a canonical or noncanonical perspective. As in Experiment 1, participants were shown a single object in the center of the screen and were asked to press a key to categorize it into one of 10 entry-level categories. The image remained on the screen until participants made their response. Participants completed sixteen practice trials before beginning the main experiment. All trial conditions were randomly interleaved. The main experiment consisted of 20 trials per condition.

## Dependent measures and analysis

### Comparison of shape and texture sensitivity with and without 3D information

We compared networks' texture bias with 2D vs. 3D images by comparing the proportion of trials in which the sum of subcategory probabilities was greatest for the object's shape with proportion of trials in which the sum of probabilities was greatest for the object's texture (see **Method**).

We compared network performance with human performance in Experiment 1 with a two-way repeated measures ANOVA. We tested for a main effect for both recognition cue and the presence of 3D information. We compared the proportion of correct trials when participants were tasked with identifying the object's shape with the proportion of correct trials when they were tasked with identifying the object's texture in Experiment 1. We compared response time for shape identification trials with response time for texture identification trials. We also tested for an effect of the inclusion of 3D shape information on participants' response accuracy and response time and for interactions between the identification task and presence of three-dimensional cues.

We hypothesized that human participants would be faster and more accurate when cued to recognize the object by its shape than its texture. We predicted that the shape advantage in human performance would be smaller or not observed at all in DNNs, which are thought to be more texture-biased than humans. When 3D information was included, we expected humans' performance on the shape recognition task to be slightly improved and their performance on the texture recognition task to remain the same or get slightly worse. We expected that the inclusion of 3D information would have a larger effect on DNNs. We predicted that the proportion of shape-based classifications to texture-based classifications would be significantly greater when 3D information was included than when an object's shape was defined only by its external contour.

## 2D vs. 3D object recognition from canonical and noncanonical perspectives

We analyzed the human data in Experiment 2 using a 2 (inclusion of 3D information) x 2 (viewpoint canonicality) repeated measures ANOVA and compared this with the proportion of correct shape classifications by DNNs from canonical and noncanonical perspectives and with or without 3D information. We predicted, as in Experiment 1, that both human and DNN performance would be better when 3D information was included.

We also hypothesized that if shading and attached shadows improve recognition by facilitating the formation of structural shape descriptions, then they should be of greater benefit to recognition of objects viewed from noncanonical perspectives than canonical perspectives. Systems that classify images based on image-level similarity would benefit from 3D information equally or more for images in which the object is seen from a canonical perspective.

## Results

### Shape and texture bias

The proportions of images that each DNN correctly classified by its shape and its texture are shown in Fig 3. We also calculated each network's texture bias with and without 3D information included. Following Geirhos et al. [35], we defined texture bias as the proportion of correct texture classifications divided by the proportion of correct shape or texture classifications. Values greater than 50% indicate a texture bias and values less than 50% indicate a shape bias.

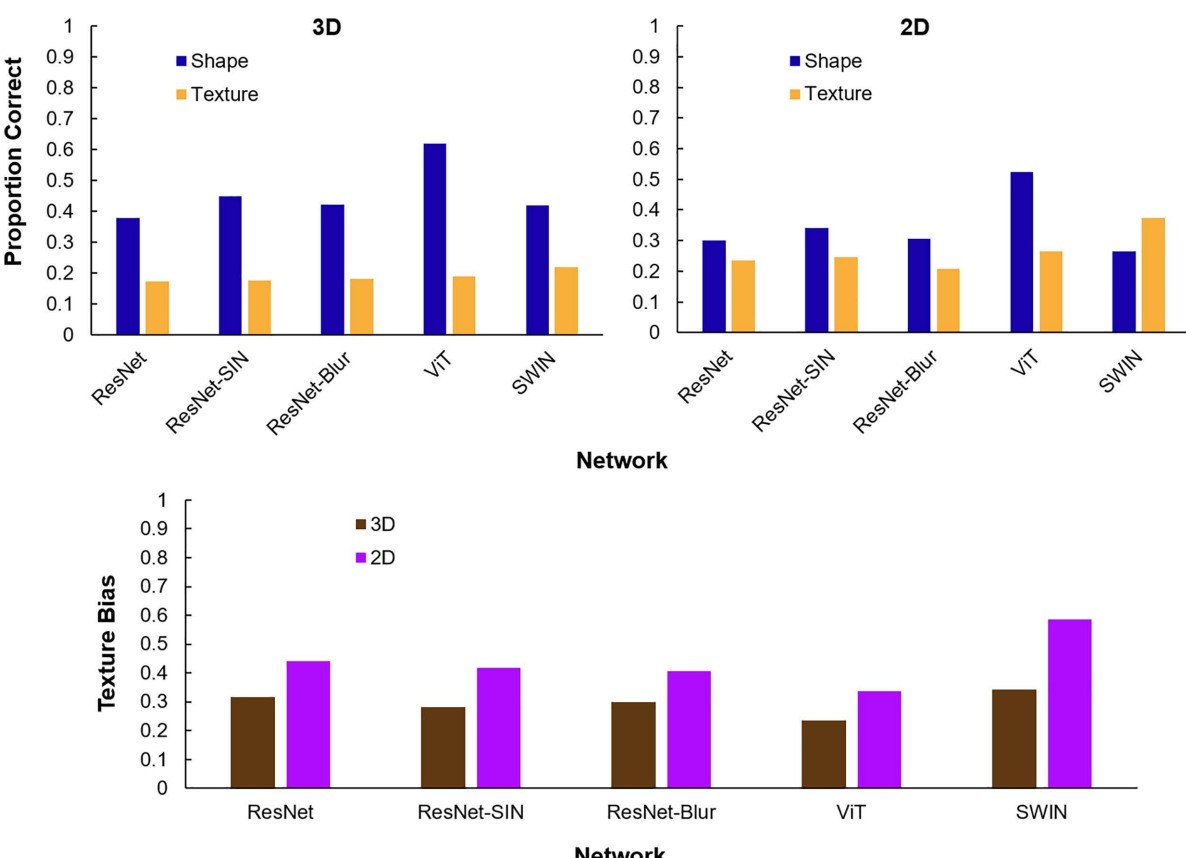

**Fig 3. Neural network shape vs. texture results.** Top row: Proportion of correct shape and texture classifications by each network with 3D and 2D information. Bottom row: Neural network texture bias in 3D and 2D, calculated as $\frac{\text{correct texture classifications}}{\text{correct texture + shape classifications}}$.

When 3D information was included, every network classified more images by their shape than by their texture. When shading and attached shadows were omitted (2D), this reversed for SWIN, and networks' shape bias was reduced in all other networks. On average, networks' shape classification accuracy was 11% better with shading and attached shadows included (46% vs. 35%) and their texture classification accuracy was 8% poorer with shading and attached shadows included (19% vs. 27%).

We also compared humans' recognition of texture and shape. The results are shown in Fig. 4. A 2x2 repeated measures ANOVA confirmed that humans responded significantly more accurately when tasked with classifying the object by its shape than by its texture ($F(1,122) = 1780.1$, $p < .001$, $\eta^2_{partial} = .94$). They also responded more accurately when 3D information was included in shapes ($F(1,122) = 179.3$, $p < .001$, $\eta^2_{partial} = .39$). Paired comparisons found that classification accuracy was significantly better for shape than texture with ($\Delta = .33$, $t(122) = 41.6$, $p < .001$, *Cohen's d* = 3.75) and without ($\Delta = .30$, $t(122) = 36.2$, $p < .001$, *Cohen's d* = 3.26) 3D information.

We also found a significant interaction between classification cue and the inclusion of 3D information ($F(1,122) = 13.4$, $p = <.001$, $\eta^2_{partial} = .10$). Participants recognized objects by their shape significantly more accurately with 3D information included ($\Delta = .042$, $t(122) = 10.57$, $p < .001$, *Cohen's d* = 0.95). Participants categorized textures more accurately with 3D information included, although this difference was smaller ($\Delta = .019$, $t(122) = 3.55$, $p = <.001$, *Cohen's d* = .32).

## Recognition of objects from canonical and noncanonical perspectives

We compared neural networks' shape classification accuracy for objects viewed from a canonical perspective and a noncanonical perspective with and without 3D information in the image. The results are displayed in Fig 5. As reported above, shape classification performance improved with 3D information included. This improvement was no greater for images viewed from a noncanonical perspective than images viewed from a canonical perspective. In fact, all five networks improved more when 3D information was included in images taken from a canonical viewpoint than when it was included in images taken from a noncanonical viewpoint.

We also analyzed the human data from Experiment 2 where viewpoint and 3D information were manipulated. The results are shown in Fig 6. A 2x2 repeated measures ANOVA confirmed significant main effects for viewpoint canonicality ($F(1,128) = 129.5$, $p < .001$, $\eta^2_{partial} = .50$) and for the inclusion of 3D vs. 2D information ($F(1,128) = 127.3$, $p < .001$, $\eta^2_{partial} = .50$).

We also found a significant interaction between viewpoint canonicality and the inclusion of 3D information ($F(1,128) = 85.3$, $p < .001$, $\eta^2_{partial} = .40$). The inclusion of 3D shape information did not significantly affect humans'

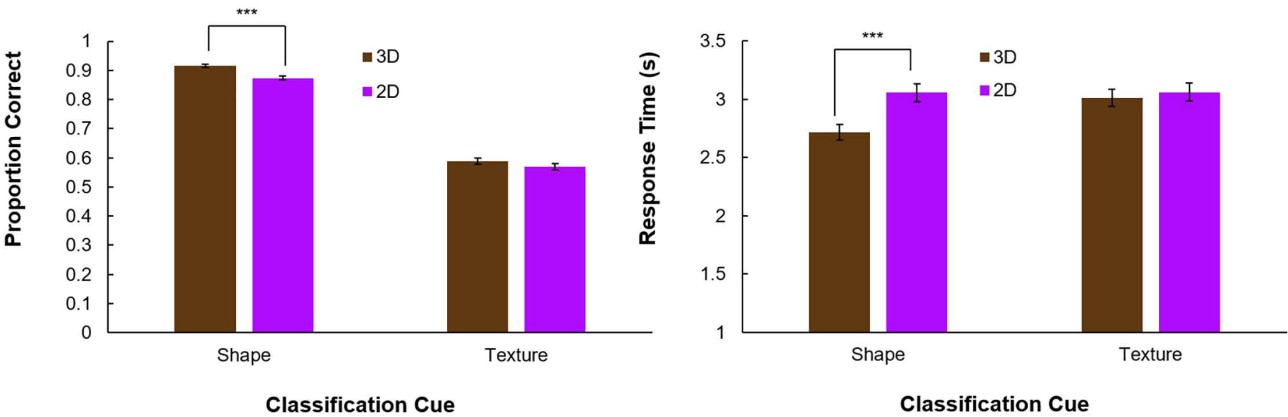

**Fig 4. Human shape vs. texture results.** Left: Classification accuracy for shape and texture classification tasks in 3D and 2D. Right: Mean response time for shape and texture classification tasks. Error bars reflect the standard errors of the mean.

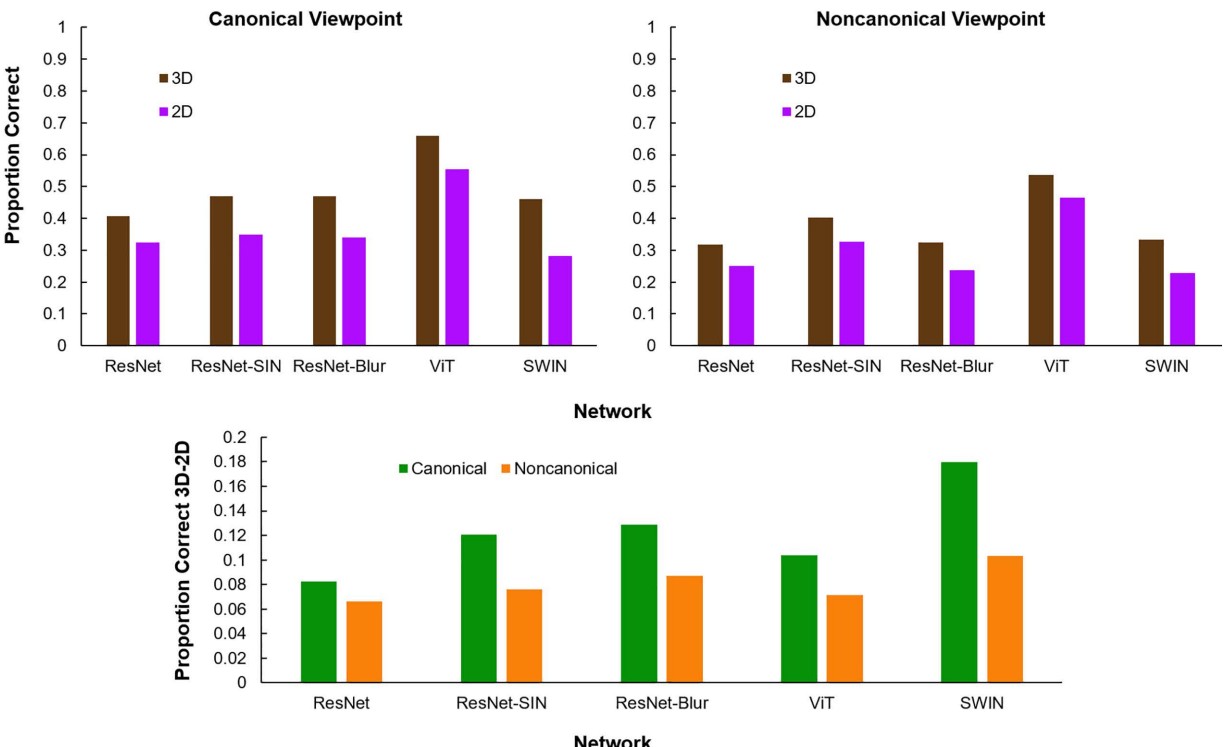

**Fig 5. Neural network canonical vs. noncanonical shape recognition results.** Top row: Network classification accuracy for images with 3D or 2D information viewed from a canonical or noncanonical perspective. Bottom row: Performance gain from the addition of 3D information in canonical and noncanonical images.

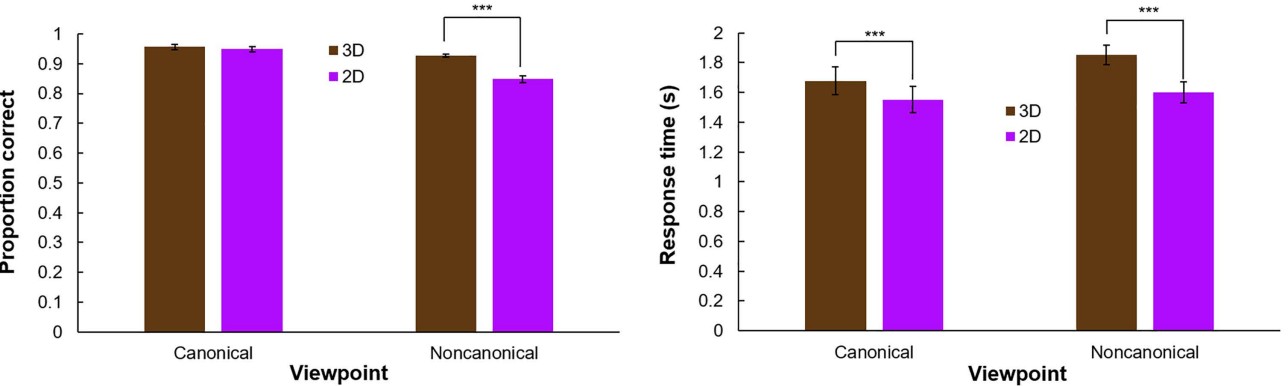

**Fig 6. Human canonical vs. noncanonical shape recognition results.** Left: Mean accuracy for images viewed from canonical vs. noncanonical perspectives with and without 3D information. Right: Mean response time. Error bars reflect the standard error of the means.

response accuracy for images viewed from a canonical perspective ($t$(128) = 1.69, $p$ = .09, *Cohen's d* = 0.15), but it did significantly improve human performance for images viewed from a noncanonical perspective ($t$(128) = 12.4, $p$ < .001, *Cohen's d* = 1.09).

### Does finetuning make networks more humanlike?

In our comparisons between models trained for ImageNet classification and human object recognition, we found that humans were substantially more shape-biased than deep networks. There are at least two interpretations of these findings. One possibility is that humans have a unique computational capacity to perceive and make use of shape for object recognition. Another possibility is that deep networks extract sufficiently robust representations of shape but weight their evidence less heavily than humans relative to texture.

One way of testing between these possibilities is by training networks on classification tasks where texture is absent or not diagnostic. Studies deploying these techniques have found that networks trained with such curricula are capable of classifying images by object shape and reducing their texture bias [35,39,64,65].

Using the previously described dataset of texture-substituted images, we finetuned ViT, the top-performing and most humanlike model from the previous experiment, to classify objects by their shape alone. One of the purposes of this experiment was to replicate earlier findings that shape bias could be greatly increased in deep networks with targeted training. We also tested whether networks finetuned to classify shapes with 3D cues present could recognize objects by only their external contour and whether networks finetuned to classify only objects presented from canonical viewpoints could also classify objects from noncanonical viewpoints with and without 3D cues. We compared all of these results with humans' performance in Experiments 1 and 2.

## Network finetuning method

### Network

All finetuning was conducted on ViT. The model was selected based on both its good performance in previous experiments and other research showing it to represent the state-of-the-art among commonly available object recognition models for modeling human performance [42,66]. We started with a model identical to the one used in the previous experiment, a model trained on JFT-300M and finetuned on ImageNet. We further finetuned it to classify between the 10 object categories in our image dataset.

### Training curricula

All training images were sourced from the 120,000 images used in testing in the previous experiment. Training was restricted to images with 3D shading cues included. We finetuned ViT using four distinct curricula:

1. **Canonical and noncanonical images with all connection weights unfrozen**: We trained the network on all possible objects across a range of viewpoints and textures. By leaving all connection weights unfrozen, we allowed the deep network to learn new features that may be pertinent to shape classification.

2. **Canonical and noncanonical images with only the connections weights between the final two layers unfrozen**: This curriculum was identical to Curriculum 1, except that only the connection weights in the final decision layer were allowed to update. Comparisons between Condition 1 and Condition 2 offer insight into which features were already extracted by ViT trained in the standard way but whose weight in the final decision needed to be updated to optimize for image classification with nondiagnostic textures.

3. **Canonical images with all connection weights unfrozen**: We also finetuned ViT using only images of objects taken from canonical viewpoints. This may simulate humans' visual diet more accurately, as we see objects from

noncanonical viewpoints much less than from canonical viewpoints [61]. In total, we had 80,400 canonical images. As in Curriculum 1, we allowed all weights to update.

4. **Canonical images with only the connection weights between the final two layers unfrozen**: We used the training images from Curriculum 3 and the procedure from Curriculum 2, allowing only the connection weights between the last and second to last layer to update.

### Training parameters

We trained ViT with a batch size of 96. We used a learning rate of $2*10^{-5}$ in Curricula 1 and 3 where all connection weights were unfrozen and a learning rate of $1*10^{-3}$ in Curricula 2 and 4 where only the decision layer's connection weights were permitted to update. We trained on 80% of the image database, withholding 20% for the validation set. We used a variable number of training epochs, stopping when error rate increased on the validation set and adopting the model with the lowest error rate on the validation set.

### Post-training comparisons

After finetuning, we tested the networks' classification accuracy on all 3D and 2D images, separating them into those shown from a canonical perspect (80,400 images) and those shown from a noncanonical perspective (39,600 images). We compared each of the finetuned models' performance with human performance in Experiment 1 and 2.

## Results

Both networks with unfrozen connection weights trained to criterion after one epoch. The networks with frozen connection weights trained to criterion after two epochs. All four networks performed well on the withheld validation set (Curriculum 1: 98.7%, Curriculum 2: 99.7%, Curriculum 3: 97.2%, Curriculum 4: 97.3%).

Fig 7 shows networks' performance on the full set of 3D and 2D images, separating canonical and noncanonical viewpoints. When ViT was finetuned with unfrozen connection weights, it performed extremely well (better than 97%) in all conditions. We found little difference between performance with the 3D images, which were also used to train the network, and 2D images, where shading and internal shadows were removed. There was also not a meaningful difference between classification accuracy for images viewed from a canonical vs. noncanonical perspective. This is especially impressive in the condition in which we finetuned ViT with only objects viewed from a canonical perspective. Despite not being directly trained on shapes viewed from a noncanonical perspective, the finetuned network classified noncanonical images very accurately (3D noncanonical: 99.3%, 2D noncanonical: 97.2%).

When we froze all but the final set of connection weights, the finetuned ViT model classified images with 3D information substantially more accurately than 2D images (97.7% vs. 83.4% when trained on all images; 95.1% vs. 81.8% when trained on only canonical images). We found no difference in classification accuracy between canonical and noncanonical images when ViT was trained on all images with frozen connection weights (3D canonical: 97.7%, 3D noncaonical: 97.7%; 2D canonical: 84.4%, 2D noncaonical: 83.2%). However, when we trained with only canonical images using frozen connection weights, we did see a meaningful difference between classification accuracy for canonical vs. noncanonical images (3D canonical: 97.8%, 3D noncanonical: 92.4%; 2D canonical: 85.6%, 2D canonical: 77.9%). Like humans, this difference was larger for 2D images (Δ=7.7%) than for 3D images (Δ=5.4%).

## Discussion

Deep networks are known to be more texture-biased than humans [33–39]. However, previous work has not systematically manipulated the inclusion of 3D shape cues such as shading and attached shadows to test their effect on object

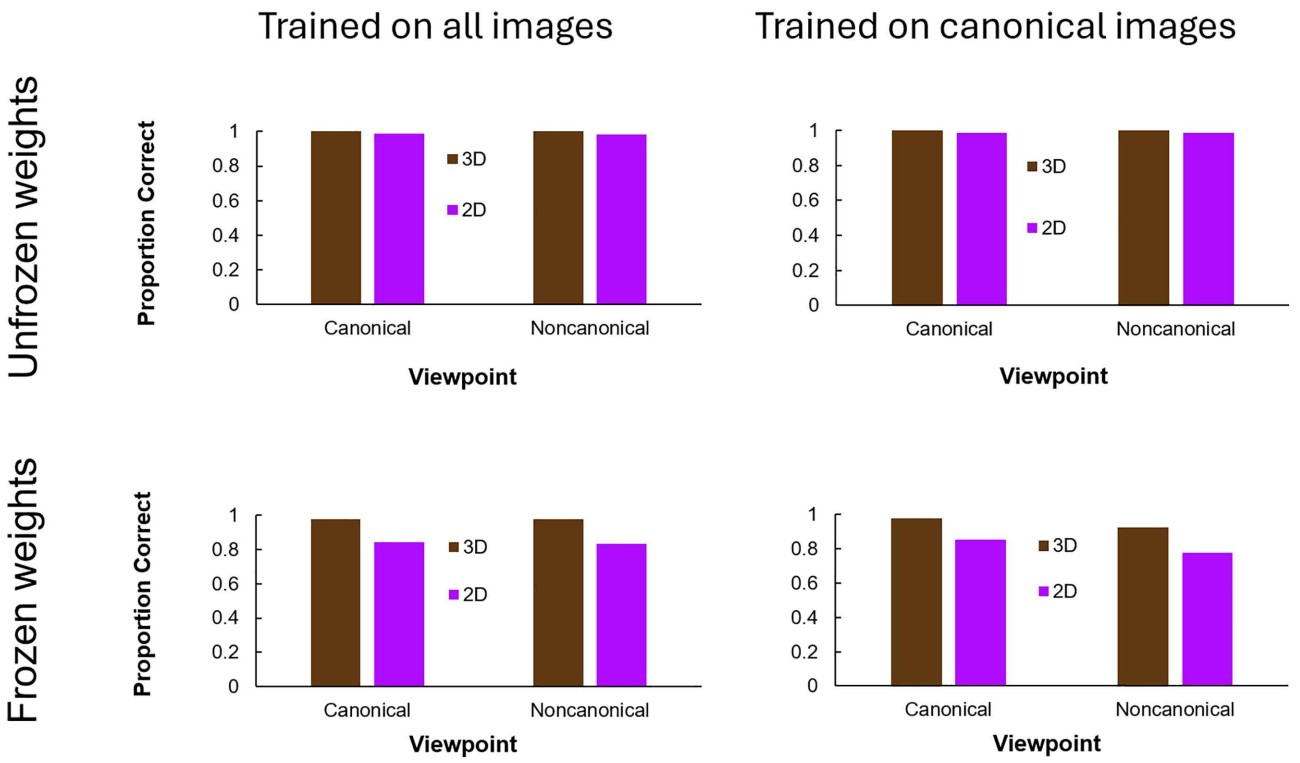

**Fig 7. ViT performance after finetuning.** Proportion of correct responses for 2D and 3D images viewed from a canonical and noncanonical perspective.

recognition and their interaction with texture cues. We designed a novel testing set using 3D models from ShapeNet to measure the effect of including 3D cues on networks' shape bias.

Consistent with previous work, we found that, absent 3D cues from the object's shape, neural networks are considerably more texture-biased than humans. In five top-performing networks, DNNs had an average texture-bias of 44% with only the external contour present. Networks classified objects correctly by their shape in an average of 35% of trials and by texture in 27% of trials. Humans classified objects correctly by shape in 87% of trials and by texture in 56% of trials without 3D cues.

The tests we used for neural networks and humans were not identical: In the neural network test, we presented a cue-conflict image and measured the probability the network assigned to both the shape and texture cue. In the human test, we explicitly instructed participants to classify images by shape or texture. These task differences cannot account for humans' superior shape recognition performance with cue-conflict stimuli. Humans' shape classification accuracy (87%) exceeded neural networks' classification accuracy for both shape and texture (61%). This held true across all networks, so even if all the classifications that neural networks made for the object's texture were counted as a correct shape classification, no DNN would have performed as well as humans on the cue-conflict stimuli.

These results show a clear difference between how information is weighted in humans' and deep networks' classification decisions. One possibility is that deep networks serve as a model of the ventral visual pathway specifically, not general object recognition [67,68]. Recently, it has been argued that global shape representations are primarily encoded in the dorsal visual pathway and integrated with more local features from the ventral visual pathway for object recognition [69]. Our findings do not speak to where in the visual brain shape is encoded, but they do align with previous research

showing its primacy in core object recognition [22,28]. Humans recognized objects much more accurately and rapidly when instructed to consider shape cues. While they could recognize objects by their texture cues significantly better than chance, this process was both slower and less reliable.

Another possibility is that humans' bias towards shape is a result of the kind of object recognition we tasked them with doing. We asked participants to recognize objects in different basic-level categories [70]. At this level of discrimination, humans rely primarily on shape, but they consider texture more for subordinate-level recognition [71–73]. Texture also plays a much larger role in specialized tasks that require training, such as the analysis of medical images [74, 75]. One reason why deep networks' texture bias may be greater than humans' is that they are trained to make both basic-level and subordinate-level discriminations between objects. For example, they need to classify both between a stingray and a hen and between a silky terrier and a soft-coated wheaten terrier.

While DNNs' texture bias is greater than humans', most networks were biased more towards shape cues than texture cues. All networks but SWIN classified more objects correctly by their shape than by their texture, even when 3D cues were omitted from the cue-conflict images. These findings challenge the view that contemporary DNNs rely primarily on texture for object recognition [34,35]. While DNNs are much more influenced by texture than humans, shape still plays an at least equal role in classification decisions.

Comparing between DNNs with different architectures or training curricula, we found surprisingly small differences in texture-bias for ResNet trained only on ImageNet and ResNet trained to specifically reduce texture-bias, such as by augmenting ImageNet with stylized [35] or blurred [39] images. With only the external contour, ResNet trained on ImageNet had a texture bias of 44%, which went down to 42% and 41% for ResNet-SIN and ResNet-Blur, respectively. With 3D shape cues included, ResNet had a texture-bias of 31%, which went down to 28% and 30% in ResNet-SIN and ResNet-Blur. These differences may be meaningful, but they are relatively modest.

ViT was the least texture-biased network among the five we tested. Intriguingly, SWIN, the other tested transformer network, was the most texture-biased. One way ViT differs from SWIN is that it has a global self-attention mechanism that allows it to learn long-range relations between pixels. This could help the network learn diagnostic features that go beyond texture, which is locally defined. SWIN, whose self-attention mechanism is constrained to learn pixel relations within a local window, may be driven to learn more local cues. Another difference between ViT and SWIN is in the quantity of data upon which each network is trained. ViT was exposed to around 300 times more images than SWIN during training. Size of training data is a major factor in other global shape cues like contour integration [18], so it could be that the differences in ViT and SWIN's training curricula, not their architectures, precipitated differences in their texture-biases.

All networks' shape bias increased substantially when 3D cues like shading and attached shadows were included from the shape model. On average, DNNs' shape bias increased from 56% to 70% with 3D cues included. Their shape recognition accuracy increased from 35% to 45%. Stimuli that put shape and texture in conflict with each other have not typically included 3D shape information because it is difficult to render separately from texture when starting from a photograph. By rendering 3D models, we preserved shapes' shading and attached shadows while substituting all other surface properties with texture from another object.

Humans also benefitted from the inclusion of 3D cues, although not by as much. Shape recognition performance increased from 87% to 92% with the addition of 3D cues. Part of humans' smaller performance gain is likely due to their excellent ability to recognize shapes with only the bounding contour [42,76,77].

We next compared the way in which 3D cues benefit recognition in biological and artificial systems. One reason that 3D cues might be beneficial is that they operate as another textural or image-level cue that is consistent with the object's shape. Images in the testing set were more similar to images that networks were trained on, or that humans had previously seen, when they included 3D cues. Luminance differences caused by shading and attached shadows might function no differently than luminance differences caused by patterns of fur or differently colored feathers. Another possibility is that humans and/or deep networks make use of shading and attached shadows to infer a shape's 3D structure. Under this

hypothesis, shadows would not be beneficial to recognition because they increase the image-level similarity between a test image and a training image, but because they increase the probability that the structural representation formed from the test image matches representations formed from previous visual experiences.

In human perception, shading, shadows, and internal contours help most with the recognition of objects viewed from a noncanonical perspective [51–53,55]. For these images, there may not be enough information in the external contour to form a structural description of an object's 3D shape [54]. Shading and attached shadows furnish additional information that aids with the formation of a 3D shape representation [45,46,78].

The human data in Experiment 2 replicated these previous findings. Humans performed similarly with and without 3D shape cues when objects were viewed from a canonical perspective (95% vs. 95%--no significant difference), but when objects were viewed from a noncanonical perspective, performance was significantly worse for images that did not include 3D cues than images that did (85% vs. 94%).

The same was not true in deep networks. On average, DNNs' accuracy was 12% higher when 3D information was included in images viewed from a canonical perspective and only 8% higher when it was included in images viewed from a noncanonical perspective. For deep networks, shading and attached shadows seem to be beneficial for recognition only because they serve as another image-level cue for recognition. Objects viewed from a canonical perspective likely benefit from these 3D cues more because there are more images including those cues from that perspective in the training data.

We further compared humans and deep networks by finetuning ViT, the top-performing network from our initial experiments. We finetuned networks with all connection weights frozen apart from those in the final decision layer and with unfrozen connection weights. Freezing all but the connection weights in the final decision layer tests whether existing features extracted by the training network can be reweighted to optimize for a different visual task. Allowing all connection weights to update tests whether new features can be learned to optimize for the new task. We also varied networks' training data, finetuning ViT both with all (i.e., both canonical and noncanonical) images and with only canonical images. Training with all images offers insight into whether sufficient information is available in an image for accurate classification while training with only canonical images more closely simulates humans' visual diets.

The results of our finetuning experiments confirmed that networks' texture biases can be dramatically reduced with targeted training. The proportion of ViT's correct classification of an object by its 3D shape rose from 62% before training to over 97% after each of the four finetuning trainings. This includes two trainings with connection weights frozen, meaning that ViT was already extracting diagnostic shape information from 3D images and could learn to prioritize shape cues over texture cues to achieve better performance in a recognition task where texture carried no diagnostic value. These findings are consistent with past research which has shown that networks trained on images with limited diagnostic texture information become substantially more shape-biased [35,39,64,65].

Deep networks finetuned to classify 3D images from both canonical and noncanonical perspectives could very accurately classify 2D images from both canonical and noncanonical perspectives as well. These results differ from humans, for whom classification of noncanonical images was significantly more accurate when 3D information like shading was also included.

To try to make networks more humanlike, we also finetuned them with only images viewed from a canonical viewpoint. This might more closely resemble humans' visual diets [61]. We found that when networks' connection weights were unfrozen, they could accurately classify both 2D and 3D noncanonical images, even when finetuned only on canonical 3D images. This is an unusual case of networks outperforming humans on a shape task. While humans benefited from structural, 3D information to reconcile shapes viewed from noncanonical viewpoints with internal representations of the shape, finetuned networks are capable of doing this with only the object's bounding contour. Possibly, DNNs learned a viewpoint invariant shape feature that supported recognition of noncanonical images. DNNs may also have learned to discount all information apart from the external contour during finetuning because of its limited diagnostic value in the curricula we administered.

The currilum that created the most humanlike behavior in ViT involved finetuning on canonical images while freezing all connection weights except the final decision layer. When finetuned in this way, ViT shared humans' slightly better performance for 3D shapes over 2D shapes (Humans: 92% vs. 87%; ViT: 95% vs. 84%). Finetuning on canonical images with frozen connection weights also resulted in a similar interaction to what was observed in humans between canonicality and 3D information. Like humans, the difference between finetuned ViT's classification accuracy for canonical vs. noncanonical images was greater for 2D images (ViT: 7.7%; Humans: 9.6%) than for 3D images (ViT: 5.4%; Humans: 1.2%), although the interaction was stronger in humans.

Whether networks finetuned in this way are really more humanlike remains uncertain. While we found a training regime that made ViT's performance align with humans more closely, humans do not need any targeted training to develop this pattern of performance. Often, when networks are trained to become more humanlike in a specific visual task, they learn to do so but this does not generalize to other visual tasks [15,41].

Our findings are consistent with a growing literature showing differences between humans' and DNNs' use of shape for object recognition. As the current work and other studies show, DNNs can classify objects based only on shape information [34,42,79]. However, research suggests that shape-based classification in neural networks relies on local shape features, not more configural aspects of shape [34,41,42,19, 80–82]. The formation of a structural, volumetric representation of object shape and subsequent use of this representation for recognition would likely require a level of abstract and configural processing beyond the capabilities of current DNN models.

An alternative hypothesis is that neither humans nor neural networks form structural representations of an object's shape. Possibly, humans benefit more from shading and attached shadows in noncanonical images because the object's external contour is more ambiguous in these images and image-level shading information is more discriminative. The pattern of results from our finetuning simulations suggest that this is not the case. DNNs serve as an ideal observer model for recognition based on image-level similarities. If shading and attached shadows were more discriminative for noncanonical images, then we should have observed performance differences between 3D and 2D noncanonical images when we finetuned ViT with unfrozen connection weights. Instead, we found that the finetuned network classified 2D and 3D images equally accurately, suggesting that shading is not simply more informative for noncanonically viewed objects at the image-level.

## Conclusion

Like many previous studies, the current work found that neural networks are substantially more texture-biased than humans. However, this texture bias is reduced when shape cues like shading and attached shadows are included in an image along with the object's external contour. It can be further reduced by finetuning on stimuli where texture is nondiagnostic. While both humans and DNNs benefit from the inclusion of 3D shape cues, they are not alike in the way these cues benefit object recognition. Human performance primarily improves for images viewed from a noncanonical perspective where the external contour may be insufficient to accurately represent the object's 3D structure, while DNNs benefit most for images of objects from familiar viewpoints where they may have had more exposure to image-level luminance patterns. These results suggest that shading and attached shadows help humans form structural, three-dimensional representations of an object while they help DNNs only with classifying objects according to image-level similarities.

## Author contributions

**Conceptualization:** Mikayla Cutler, Luke Baumel, George K. Thiruvathukal, Nicholas Baker.

**Data curation:** Mikayla Cutler, Luke Baumel, Joseph Tocco, William Friebel, Nicholas Baker.

**Formal analysis:** Luke Baumel, William Friebel, Nicholas Baker.

**Funding acquisition:** Luke Baumel.

**Investigation:** Mikayla Cutler, Luke Baumel, Joseph Tocco, William Friebel, Nicholas Baker.

**Methodology:** Mikayla Cutler, Luke Baumel, Joseph Tocco, George K. Thiruvathukal, Nicholas Baker.

**Project administration:** Nicholas Baker.

**Software:** Mikayla Cutler, Luke Baumel, George K. Thiruvathukal, Nicholas Baker.

**Supervision:** George K. Thiruvathukal, Nicholas Baker.

**Visualization:** Mikayla Cutler.

**Writing – original draft:** Mikayla Cutler, Luke Baumel, William Friebel, George K. Thiruvathukal, Nicholas Baker.

**Writing – review & editing:** Mikayla Cutler, William Friebel, George K. Thiruvathukal, Nicholas Baker.

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
