## [Decision Letter · Decision Letter 0]

29 Jan 2026

PONE-D-25-61879Three-dimensional shape cues affect human and artificial recognition systems differentlyPLOS One

Dear Dr. Baker,

Thank you for submitting your manuscript to PLOS ONE. After careful consideration, we feel that it has merit but does not fully meet PLOS ONE’s publication criteria as it currently stands. Therefore, we invite you to submit a revised version of the manuscript that addresses the points raised during the review process.

We look forward to receiving your revised manuscript.

Kind regards,

Debotosh Bhattacharjee, PhD

Academic Editor

PLOS One

Journal Requirements:

2. Please note that PLOS One has specific guidelines on code sharing for submissions in which author-generated code underpins the findings in the manuscript. In these cases, we expect all author-generated code to be made available without restrictions upon publication of the work.

Please review our guidelines at https://journals.plos.org/plosone/s/materials-and-software-sharing#loc-sharing-code and ensure that your code is shared in a way that follows best practice and facilitates reproducibility and reuse.

“This work was funded by a Carbon Fellowship to LB (https://www.luc.edu/sustainability/research/studentresearchopportunities/)”

6. We note that Figures 1 and 2 in your submission contain copyrighted images. All PLOS content is published under the Creative Commons Attribution License (CC BY 4.0), which means that the manuscript, images, and Supporting Information files will be freely available online, and any third party is permitted to access, download, copy, distribute, and use these materials in any way, even commercially, with proper attribution. For more information, see our copyright guidelines: http://journals.plos.org/plosone/s/licenses-and-copyright.

1) You may seek permission from the original copyright holder of Figures 1 and 2 to publish the content specifically under the CC BY 4.0 license.

2) If you are unable to obtain permission from the original copyright holder to publish these figures under the CC BY 4.0 license or if the copyright holder’s requirements are incompatible with the CC BY 4.0 license, please either i) remove the figure or ii) supply a replacement figure that complies with the CC BY 4.0 license. Please check copyright information on all replacement figures and update the figure caption with source information.

If applicable, please specify in the figure caption text when a figure is similar but not identical to the original image and is therefore for illustrative purposes only.

7.  Please remove your figures from within your manuscript file, leaving only the individual TIFF/EPS image files, uploaded separately. These will be automatically included in the reviewers’ PDF.

**Additional Editor Comments:**

Try to highlight your contribution and carefully revise the paper as per review reports.

Reviewers' comments:

Reviewer's Responses to Questions

**Comments to the Author**

1. Is the manuscript technically sound, and do the data support the conclusions?

Reviewer #1: Partly

Reviewer #2: Partly

2. Has the statistical analysis been performed appropriately and rigorously? 

Reviewer #1: Yes

Reviewer #2: I Don't Know

3. Have the authors made all data underlying the findings in their manuscript fully available?

Reviewer #1: Yes

Reviewer #2: No

4. Is the manuscript presented in an intelligible fashion and written in standard English?

Reviewer #1: Yes

Reviewer #2: Yes

5. Review Comments to the Author

Reviewer #1: The statement repeatedly used in the abstract and other partsof the manuscript: “While shape is the primary cue in human object recognition…”, is not accurate from a biological and cognitive science perspective and therefore requires modification.

Extensive evidence from neuroscience and cognitive vision literature demonstrates that texture-based processing comes before shape-based dominance in the human visual system. Work by Felleman in “Distributed hierarchical processing in the primate cerebral cortex” shows that early visual areas process local texture features before global shape representations. Again, Okazawa in “Gradual Development of Visual Texture-Selective Properties Between Macaque Areas V2 and V4” states that for distributed and hierarchical processing texture information is extracted earlier than shape. Furthermore, Adelson, in “On seeing stuff: The perception of materials by humans and machines”, explicitly states that humans rely on texture early and shape later. This aligns well with the CNN-based models which are texture-biased and it makes CNN biologically similar like visual processing rather than contradictory to human cognition.

It is true that the manuscript presents strong statistical and experimental results emphasizing the importance of shape. It also highlights a misconception that “shape is the primary cue in human object recognition." Yes, it is true, as discussed by DiCarlo in “How does the brain solve visual object recognition?”, shape-based representations dominate in high-level cortical processing but are not necessarily primary in simpler or domain-specific recognition tasks, such as medical imaging or known object categories. Therefore, the manuscript would significantly need to rewrite by improving the followings:

• Texture plays a dominant role in early visual processing.

• Shape becomes increasingly important at higher cognitive levels.

• CNN texture bias does not contradict human vision but instead reflects early-stage biological processing.

• Shape dominance is context-dependent and task-specific.

Integrating the above neuroscientific and cognitive insights would make the manuscript conceptually richer, biologically more accurate, and better justified, rather than presenting the shape before texture relationship as a strict claim.

Reviewer #2: The sample size of human participants is very small, and the results are not representative enough. Give more details about the hypothesis formulations for the human participants, and state the experimental setup with more details.

As modern deep learning techniques outperform human experts in object classification and texture analysis. While modern data augmentation can also transform the (non-)canonical objects more occluded and challenging via different transformations. Therefore, this method is not convincing to me.

Further, more experimental description is required to reach the conclusion. For example, how the performance of a deep model can be if cross validation is applied ? or, train the model from scratch and validate the performance. There are also some standard methods for addressing the bias and overfitting issues, which should also be tested.

6. PLOS authors have the option to publish the peer review history of their article (what does this mean?). If published, this will include your full peer review and any attached files.

Reviewer #1: **Yes:** Hiranmoy Roy

Reviewer #2: No

---

## [Author Response · Author response to Decision Letter 1]

14 Apr 2026

Changes addressing the reviewers’ concerns are written in red text in the revised submission for the reviewers’ convenience. Line numbers are referenced in our response to the reviewers.

Reviewer #1:

The statement repeatedly used in the abstract and other parts of the manuscript: “While shape is the primary cue in human object recognition…”, is not accurate from a biological and cognitive science perspective and therefore requires modification.

Extensive evidence from neuroscience and cognitive vision literature demonstrates that texture-based processing comes before shape-based dominance in the human visual system. Work by Felleman in “Distributed hierarchical processing in the primate cerebral cortex” shows that early visual areas process local texture features before global shape representations. Again, Okazawa in “Gradual Development of Visual Texture-Selective Properties Between Macaque Areas V2 and V4” states that for distributed and hierarchical processing texture information is extracted earlier than shape. Furthermore, Adelson, in “On seeing stuff: The perception of materials by humans and machines”, explicitly states that humans rely on texture early and shape later. This aligns well with the CNN-based models which are texture-biased and it makes CNN biologically similar like visual processing rather than contradictory to human cognition.

We thank the reviewer for pointing out the ways we oversimplified shape vs. texture use in human visual object recognition. As the reviewer points out, the role of texture is not merely secondary to shape but serves many complementary purposes. We have updated sections of the revised submission to present a more thorough analysis of humans’ use of shape and texture for object recognition. In the Abstract, we have revised lines 25-26 to emphasize the importance of shape in human object recognition without directly placing it above other cues like texture. We have also substantially edited the second paragraph of the Introduction (lines 67-86) to discuss the separate roles and dynamics between shape and texture processing.

The reviewer’s comment regarding texture being processed before shape in the ventral visual stream is particularly intriguing. In addition to the references mentioned by the reviewer, a recent paper by Wright, Dering, Martinovic & Gheorghiu (2020) suggests that texture is processed before shape but that the goal of early texture processing is to de-texturize the image to aid shape-based classification. Other papers (Banno & Saiki, 2015 & Jagadesh & Gardner, 2022) suggest that texture could serve as a powerful cue for “ultrarapid” object recognition. These nuances are important and we feel the manuscript is significantly strengthened by engaging with them more thoroughly.

It is true that the manuscript presents strong statistical and experimental results emphasizing the importance of shape. It also highlights a misconception that “shape is the primary cue in human object recognition." Yes, it is true, as discussed by DiCarlo in “How does the brain solve visual object recognition?”, shape-based representations dominate in high-level cortical processing but are not necessarily primary in simpler or domain-specific recognition tasks, such as medical imaging or known object categories.

We agree with the reviewer that humans’ texture vs. shape consideration depends greatly on the task they are performing and this was oversimplified in the previous submission. For this article, our focus was on “basic-level” object recognition, or the recognition of objects commonly named different things in natural language. Texture seems to play a larger role in subordinate-level object recognition in humans as well as in specialized image analysis like medical images. We have added a paragraph discussing this in the Discussion (lines 532-541) and conjecture that this could be one reason for DNNs’ greater texture bias.

Therefore, the manuscript would significantly need to rewrite by improving the followings:

• Texture plays a dominant role in early visual processing.

• Shape becomes increasingly important at higher cognitive levels.

• CNN texture bias does not contradict human vision but instead reflects early-stage biological processing.

• Shape dominance is context-dependent and task-specific.

Integrating the above neuroscientific and cognitive insights would make the manuscript conceptually richer, biologically more accurate, and better justified, rather than presenting the shape before texture relationship as a strict claim.

In addition to the changes described above, we have added a paragraph to the Discussion (lines 521-531) discussing the possibility that DNNs are models not of object recognition overall but specifically of the ventral visual stream. Recent research posits that the VVS primarily identifies local shape and texture features and that these are integrated into global shape representations through lateral connections with the dorsal visual stream. In the revision, we try to make clear that the differences we identified between humans and DNNs are in reference to basic-level visual object recognition on the whole, not to DNNs’ validity as models of specific brain regions.

We hope that the reviewer is satisfied with our greater engagement with the cognitive and neuroscientific literature about shape and texture processing in human perception. We thank the reviewer for their helpful suggestions and agree that discussion of these texts has strengthened the message of our article.

Reviewer #2:

The sample size of human participants is very small, and the results are not representative enough. Give more details about the hypothesis formulations for the human participants, and state the experimental setup with more details.

We recruited an additional 80 participants (125 total in Experiment 1 / 131 in Experiment 2 the revision vs. 45 in the previous submission) for the two human experiments (line 250). The additional data did not change any of our major findings, but as the reviewer says, greatly increases the representativeness of our sample.

We also added information to the revision to more clearly explain the setup (lines 257-260), procedure (lines 262-293), and hypotheses (lines 295-328) of the two experiments we conducted.

As modern deep learning techniques outperform human experts in object classification and texture analysis. While modern data augmentation can also transform the (non-)canonical objects more occluded and challenging via different transformations. Therefore, this method is not convincing to me.

Further, more experimental description is required to reach the conclusion. For example, how the performance of a deep model can be if cross validation is applied ? or, train the model from scratch and validate the performance. There are also some standard methods for addressing the bias and overfitting issues, which should also be tested.

We thank the reviewer for raising these concerns. As the reviewer says, we primarily focused our comparison between humans and deep networks on pretrained networks optimized for ImageNet classification. We consider this to be an important comparison to make because considerable research in neuroscience and cognitive science has posited that networks trained in this way model human visual perception.

However, we agree with the reviewer that the novelty and generality of our findings could be improved by training a network ourselves to test how targeted training affects shape vs. texture bias, the use of external vs. internal shape cues, and recognition of objects from canonical and noncanonical viewpoints.

To this end, we finetuned the best-performing model from the previously reported experiments (ViT) and finetuned it using four distinct training regimes to test whether shape vs. texture bias can be altered by targeting training and whether this kind of training affects a finetuned networks’ ability to recognize objects from canonical and noncanonical viewpoints with or without 3D information.

The experiment and results have been added to lines 401-496 of the revised manuscript. We found that finetuning on texture-substituted stimuli significantly increased ViT’s shape-bias to align more closely with humans. When finetuned with all connection weights free to update, ViT showed no difference in recognition of canonical vs. noncanonical images nor any difference between objects with 3D shape information included vs. only 2D shape information included. This held true even when we finetuned the network only with images of objects viewed from a canonical perspective. When we finetuned with all but the final decision layer’s connection weights frozen, we found that ViT benefitted significantly from the inclusion of 3D shape information. When trained with only canonical images and frozen connection weights, ViT’s performance followed many of the same patterns observed in the human data. We discuss these findings in the Discussion section in lines 611-654.

We think these finetuning experiments will address the reviewer’s concerns that some of the findings from our previous manuscript can be addressed through data augmentation or further training. As the reviewer suggested, the shape vs. texture bias previously observed can be addressed through targeted finetuning, although we note in the Discussion that humans do not need explicit exposure to images of objects with nondiagnostic texture to develop a shape bias (lines 655-659).

---

## [Decision Letter · Decision Letter 1]

4 May 2026

Three-dimensional shape cues affect human and artificial recognition systems differently

PONE-D-25-61879R1

Dear Dr. Baker,

We’re pleased to inform you that your manuscript has been judged scientifically suitable for publication and will be formally accepted for publication once it meets all outstanding technical requirements.

Kind regards,

Debotosh Bhattacharjee, PhD

Academic Editor

PLOS One

Reviewers' comments:

Reviewer's Responses to Questions

**Comments to the Author**

Reviewer #1: All comments have been addressed

Reviewer #2: All comments have been addressed

2. Is the manuscript technically sound, and do the data support the conclusions?

Reviewer #1: Yes

Reviewer #2: Yes

3. Has the statistical analysis been performed appropriately and rigorously? 

Reviewer #1: Yes

Reviewer #2: Yes

4. Have the authors made all data underlying the findings in their manuscript fully available?

Reviewer #1: Yes

Reviewer #2: Yes

5. Is the manuscript presented in an intelligible fashion and written in standard English?

Reviewer #1: Yes

Reviewer #2: Yes

6. Review Comments to the Author

Reviewer #1: The authors have addressed all my queries, and I am satisfied with their responses. Therefore, the manuscript can be accepted in its current form.

Reviewer #2: (No Response)

---

## [Editor Report · Acceptance letter]

PONE-D-25-61879R1

PLOS One

Dear Dr. Baker,

I'm pleased to inform you that your manuscript has been deemed suitable for publication in PLOS One. Congratulations! Your manuscript is now being handed over to our production team.

Kind regards,

on behalf of

Dr. Debotosh Bhattacharjee

Academic Editor

PLOS One